# PLAYBOOK: SCALABLE DISCRETE SKILL DISCOVERY FROM UNSTRUCTURED DATASETS FOR LONG-HORIZON DECISION-MAKING PROBLEMS

## ABSTRACT

Skill discovery methods equip an agent with diverse skills necessary for solving challenging tasks through an unsupervised learning manner. However, making the pre-learned skills expandable for new tasks remains a challenge in existing research. To handle this limitation, we propose a scalable skill discovery algorithm, a *playbook*, which can accommodate unseen tasks by training new skills while maintaining previously learned ones. The playbook, characterized by discrete skills and an extendable structure, enables the extension of the skill set to cover new datasets. Since we design the playbook to have a finite number of skills, we can interpret a decision-making problem as a sequential skill classification problem, so we aim to learn additional skills of the playbook by applying the techniques of class-incremental learning. In addition, we also introduce skill planning schemes that can leverage both previously and newly learned skills to solve challenging tasks compounded by multiple sub-tasks. The proposed method is evaluated in the complex robotic manipulation benchmarks, and the results show that the playbook outperforms existing state-of-the-art methods that learn continuous skills.

## 1 INTRODUCTION

Recent studies on skill discovery have successfully addressed challenging decision-making tasks such as maze navigation (Pertsch et al., 2020; Shi et al., 2022; Kim et al., 2023), locomotion (Sharma et al., 2020; Kim et al., 2021), and robotic manipulation (Ajay et al., 2021; Hong et al., 2024). Skill discovery, a hierarchical policy learning method, equips an agent with the skills necessary to solve complex tasks by identifying and acquiring useful and diverse skills through an unsupervised learning manner. These methods learn the skill space or skill set by embedding sampled trajectories from task-agnostic datasets, which are collected by actively exploring the environment (Jiang et al., 2022; Mazzaglia et al., 2023) or pre-collected using a behavior policy (Gupta et al., 2019; Lynch et al., 2019; Rosete-Beas et al., 2022). The discovered skills can be leveraged for solving downstream tasks, e.g., reaching goals or maximizing rewards designed for a specific task.

In order to apply skill discovery methods to more general and everyday tasks, the learned skill space or skill set must be scalable. For example, a cooking robot in a kitchen should be able to learn additional recipes using new cooking tools or ingredients. However, there is a lack of research on learning new skills and expanding an available skill set for unseen tasks. Existing studies such as Eysenbach et al. (2019); Lee et al. (2020); Peng et al. (2019); Laskin et al. (2022); Park et al. (2022; 2023) also solve downstream tasks using pre-acquired skills, while they often struggle to solve entirely unseen tasks because their learned skill space cannot be expanded. To address this issue, we propose a *playbook*, a novel algorithm with a scalable structure that allows us to add skills for new tasks while maintaining previously learned skills.

If the skills are discrete, we can interpret a goal-conditioned decision-making problem as a sequential skill classification problem. Also, adding skills implies increasing the number of classes the agent can select. From this perspective, the main idea of the playbook is to extend the finite skill set by applying the techniques of class-incremental learning for image classification. To do this, we design the playbook to select a skill based on current and goal states. As a result, the playbook

can be extended to accommodate new tasks by learning additional skills through class-incremental learning. To mitigate the problem of losing previous skills when learning new skills, which is called catastrophic forgetting, we utilize the gradient boosting method of Wang et al. (2022), which fixes previous skills while training new skills. By using the extended playbook, we can solve compounded problems, which are a mixture of old and new tasks. For instance, if we have a pre-trained playbook that can open a drawer and extend it for the new task of picking up a block, an extended playbook can pick up the block in the closed drawer.

The playbook focuses on training a set of discrete skills. However, it is challenging to express multi-modal behavior distributions with only a finite number of skills. To solve this issue, the playbook utilizes the MCP (Peng et al., 2019) structure. MCP has several behavior primitives, each of which represents an independent action probability distribution. MCP generates a wide range of behaviors by combining primitives with a weight vector. We let the playbook learn finite skills, which are used as weight vectors of MCP and primitives. Then, we can extend the playbook by adding new skills and primitives to increase its expressive power over the raw action space for solving unseen tasks.

Our primary contribution is to propose a scalable skill discovery method that can accommodate new tasks by expanding the discrete skill set. The playbook has the following strengths: 1) The playbook covers multi-modal behavior distributions with a small number of skills. We have experimentally verified that the playbook with a finite number of skills shows better performance than existing baselines that train the continuous skill space. Specifically, on the CALVIN benchmark (Mees et al., 2022), the playbook achieves a success rate of 21.4% for challenging robotic problems that require an agent to decide and perform several hidden tasks. The existing state-of-the-art methods hardly solve these problems (success rate of 1.3% or less). 2) The playbook can be extended to adopt new skills. We also have experimentally verified that when new datasets are provided on CALVIN, the playbook successfully addresses new tasks included in datasets through structural extension. 3) The extended playbook can solve compounded problems by mixing skills learned from different datasets. We have verified that the extended playbook records a success rate of 24.4% for challenging compounded problems on CALVIN. The source code is provided in the supplementary material.

## 2 RELATED WORK

### 2.1 HIERARCHICAL POLICY LEARNING USING AN OFFLINE TASK-AGNOSTIC DATASET

Recent research on skill discovery has successfully solved long-horizon tasks by acquiring skills from an offline task-agnostic dataset (Lynch et al., 2019; Singh et al., 2021; Hakhamaneshi et al., 2022). These studies generally aim to solve intricate tasks by deploying skills and address downstream tasks by reusing or fine-tuning previously learned skills. Existing methods learn the skill space by encoding actions (Pertsch et al., 2020), states (Gupta et al., 2019), and state-action pairs (Ajay et al., 2021; Shi et al., 2022) of the dataset and generate raw actions using a skill-conditioned policy. On the other hand, the playbook trains a set of skills by embedding a state and an action sequence and uses the skill as a weight vector of the MCP structure rather than a direct input of the policy. Skill discovery studies such as Mazzaglia et al. (2023); Ju et al. (2024) train discrete skills to represent multi-modal behaviors of offline datasets, but they do not expand the skill set for new tasks. In contrast, the playbook extends their skills and structure to cover completely new tasks.

### 2.2 CLASS-INCREMENTAL LEARNING FOR IMAGE CLASSIFICATION

Traditional supervised image classification methods (Szegedy et al., 2015; Simonyan & Zisserman, 2015; He et al., 2016) excel in static settings but face challenges in class-incremental learning due to the catastrophic forgetting problem. There exist techniques to address this issue, such as the dynamic architecture method (Rusu et al., 2016; Kirkpatrick et al., 2017), which utilizes a flexible neural network structure to accommodate new classes, and the knowledge distillation method (Hinton et al., 2015; Li & Hoiem, 2016), which transfers knowledge from a larger or pre-trained model to a smaller or evolving one. Meanwhile, FOSTER (Wang et al., 2022) applies the gradient boosting method for mitigating catastrophic forgetting of image classification. The gradient boosting method (Ke et al., 2017; Dorogush et al., 2018) minimizes an empirical error for a new dataset by iteratively adding weak functions to the existing one. FOSTER defines additional parameterized models to cover a new dataset while fixing the existing models and minimizes an image classification loss. We employ

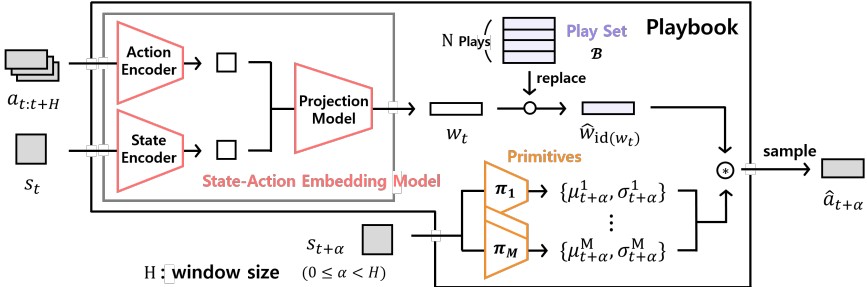

Figure 1: Overview of the structure of the playbook. The state-action embedding model embeds a state and an action sequence from an offline dataset to select one play vector among the play set. Each primitives outputs an action distribution by using a state as an input. The playbook uses the selected play vector as weights to form a single action distribution with primitives.

the gradient boosting method of FOSTER to extend the playbook to accommodate a new dataset while mitigating the catastrophic forgetting issue for a skill set.

## 3 LEARNING PLAYBOOK FROM UNSTRUCTURED DATASETS

In this section, we propose a novel scalable skill discovery method, a *playbook*, which aims to learn a finite number of skills capable of representing multi-modal behavior distributions included in unstructured datasets consisting only of states and actions without any task description. In this paper, we refer to a skill of the playbook as a *play*. As show in Figure 1, the playbook utilizes a state-action embedding model to select a play from the set of $N$ plays according to a given state and an action sequence. The selected play becomes the weights of $M$ primitives to let the weighted combination of primitives represent the given raw action sequence. We design the playbook as a structure of individual and independent components (i.e., multiple plays and primitives) to facilitate structural extension. Consequently, the playbook can improve the expressive power by increasing the number of plays and primitives it owns. Section 3 introduces the playbook structure and explains how the playbook is trained. Section 4 presents the process of extending a playbook through class-incremental learning. Finally, Section 5 describes the play plan method for reaching the given goal state using a trained playbook.

### 3.1 PLAY SELECTION THROUGH EMBEDDING MODEL

The state-action embedding model parameterized by $\theta$ maps a state and an action sequence to a play belonging to a set of $N$ play vectors. First, a state and an action sequence are encoded separately and then projected into a raw vector $w \in \mathbb{R}^M$, where $M$ is the number of primitives. Next, using the vector quantization technique (Van Den Oord et al., 2017), $w$ is replaced by the closest play $\hat{w}$ in the play set $\mathcal{B} = \{\hat{w}_1, \cdots, \hat{w}_N\} \subset \mathbb{R}^M_{\geq 0}$ as follows:

$$\text{quantization}(w) = \hat{w}_j \qquad j = \underset{i}{argmin} \|w - \hat{w}_i\|_2, \qquad i \in \{1, 2, \cdots, N\}. \tag{1}$$

Since we select a play in the play set, we can consider the play as a discrete variable. In other words, each play is expressed as an integer, which indicates the index within the play set. Note that the play is not used as a direct input to a skill-conditioned policy like previous studies. Instead, it serves as a weight vector to combine primitives described in the next section.

### 3.2 PLAYBOOK LEARNING WITH PRIMITIVES

The playbook utilizes an MCP structure (Peng et al., 2019) to present diverse and useful action distributions. MCP has multiple primitives, and each primitive is an independent probability distribution over the action space. In MCP, primitives are integrated with a weight vector into one action distribution, called a composite policy. There are two reasons for utilizing MCP in the playbook. First, MCP provides a more flexible range of behaviors by combining multiple primitives rather than

choosing one primitive. It is advantageous in expressing multi-modal behaviors with a finite number of plays. Second, MCP using multiple primitives is suitable for the playbook extension. Since primitives of MCP are independent, they can be added without affecting each other to improve its expressive capacity over the action space.

The playbook has $M$ parameterized primitives $\{\pi_{\phi_1}, \cdots, \pi_{\phi_M}\}$, and each primitive presents an independent probability distribution over the raw action space, $\pi_{(\cdot)}(a|s)$, taking the state $s$ as an input. Then, the playbook combines $M$ primitives into one action distribution, a composite policy $\hat{\pi}$, using a selected play. The composite policy is defined as follows:

$$\hat{\pi}(a|s,\hat{w}) = \frac{1}{Z(s,\hat{w})} \prod_{m=1}^{M} \pi_{\phi_m}(a|s)^{\hat{w}[m]}, \quad \hat{w}[m] \geq 0, \tag{2}$$

where $Z(s,\hat{w})$ is a partition function for the normalization of the composite policy, and $\hat{w}[m]$ is the $m$-th element of weight vector $\hat{w} \in \mathcal{B}$. MCP derives the composite policy as a Gaussian distribution by modeling each primitive as Gaussian. More details for the MCP formula for the playbook are presented in Appendix A. Then, we train the playbook using the following loss:

$$\mathcal{L}_{\{\theta,\mathcal{B},\phi_1,\cdots,\phi_M\}} = \mathop{\mathbb{E}}_{\tau \sim \mathcal{D}, t, w_t \sim p_\theta(w|\tau)} \Big[ -\log(\hat{\pi}(a_t|s_t, \hat{w}_{id(w_t)}))$$
$$+ c_1 \|\mathbf{sg}[w_t] - \hat{w}_{id(w_t)}\|_2^2 + c_2 \|w_t - \mathbf{sg}[\hat{w}_{id(w_t)}]\|_2^2 \Big], \tag{3}$$

where $\mathcal{D}$ is the given dataset, $\mathbf{sg}$ is the stop-gradient operator, $id(w)$ indicates the index of the play closest to the raw vector $w$, and $c_1$ and $c_2$ are constants. The first term maximizes the likelihood of the composite policy, while the other terms encourage raw vectors and plays to get close.

### 3.3 INFORMATION BOTTLENECK OBJECTIVE FOR ACTION ENCODER

Since the playbook uses a finite number of plays, the embedding model of the playbook can be considered to classify an action sequence into one play in a given state. To effectively train the embedding model, the action encoder has to find out the intentions contained in the multi-modal actions of datasets. If different action sequences have similar intentions to perform, i.e., the states to be reached are similar, they should be mapped into the same play. To this end, we aim to extract intentions from raw actions through the action encoder using the information bottleneck (IB)-based objective.

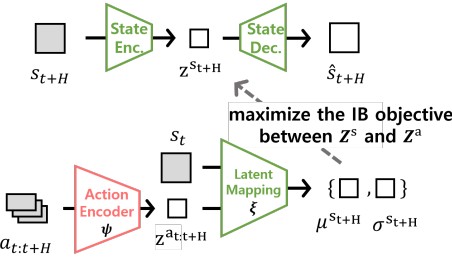

Figure 2: Additional models for extracting information from actions using the IB objective.

We propose the IB objective, which is estimated using an action encoder, a latent mapping model and a state encoder-decoder pair, as depicted in Figure 2. First, a state and an action sequence are embedded into state latent $\mathbf{z}^{s_{t+H}}$ and action latent $\mathbf{z}^{a_{t:t+H}}$ through different encoders, respectively. In particular, action latent implies an intention of an action sequence of fixed length. Next, we define the latent mapping model parameterized by $\xi$ as $p_\xi(\mathbf{z}^{s_{t+H}}|\boldsymbol{s}_t, \mathbf{z}^{a_{t:t+H}})$, which predicts the distribution of the future state latent as Gaussian using the current state and action latent as an input. The action encoder parameterized by $\psi$ and the latent mapping model parameterized by $\xi$ have the following IB objective between state and action latents:

$$\text{maximize} \quad \mathbb{E}_t \left[ \mathcal{I}(Z^{s_{t+H}}; Z^{a_{t:t+H}}|S_t) - \beta \mathcal{I}(Z^{a_{t:t+H}}; A_{t:t+H}) \right], \tag{4}$$

where $S_t$, $A_{t:t+H}$, $Z^{s_t}$, and $Z^{a_{t:t+H}}$ are random variables corresponding to $\boldsymbol{s}_t$, $\boldsymbol{a}_{t:t+H}$, $\mathbf{z}^{s_t}$, and $\mathbf{z}^{a_{t:t+H}}$, respectively, and $\beta$ is a constant. The above IB objective is interpreted as follows. The first term, $\mathcal{I}(Z^{s_{t+H}}; Z^{a_{t:t+H}}|S_t)$, means that given the state $\boldsymbol{s}_t$, an action latent $\mathbf{z}^{a_{t:t+H}}$ is informative about a state latent $\mathbf{z}^{s_{t+H}}$. Next, the second term, $\mathcal{I}(Z^{a_{t:t+H}}; A_{t:t+H})$, means that an action latent $\mathbf{z}^{a_{t:t+H}}$ is penalized for preserving information about an action sequence $\boldsymbol{a}_{t:t+H}$. That is, although $\mathbf{z}^{a_{t:t+H}}$ is extracted from raw actions, since $\mathbf{z}^{a_{t:t+H}}$ only has the minimum information to infer $\mathbf{z}^{s_{t+H}}$, the meaning of each action is lost, and the intention of the action sequence remains. We derive the following lower bound of (4), which is used as the additional loss term for the playbook:

$$\mathcal{L}_{\{\psi,\xi\}} = - \mathop{\mathbb{E}}_{\tau, t, z^{a_{t:t+H}}, z^{s_{t+H}}} \Big[ \log p_\xi(z^{s_{t+H}}|s_t, z^{a_{t:t+H}})$$
$$- \log \mathbb{E}_{z^a} \big[ p_\xi(z^{s_{t+H}}|s_t, z^a) \big] - \beta D_{KL} \big( p_\psi(Z^{a_{t:t+H}}|a_{t:t+H}) \,\|\, q(Z^{a_{t:t+H}}) \big) \Big]. \tag{5}$$

The derivation of the lower bound (5) can be found in Appendix B. On the other hand, we train the state encoder and decoder independently using the $\beta$-VAE (Higgins et al., 2017) loss, $\mathcal{L}_{VAE}$, to prevent the state encoder from falling into trivial solutions such as converging state latents to the zero vector. As a result, we train the playbook by minimizing the integrated loss $\mathcal{L} = \mathcal{L}_{\{\theta, \mathcal{B}, \phi_1, \cdots, \phi_M\}} + c_3 \mathcal{L}_{\{\psi, \xi\}} + c_4 \mathcal{L}_{VAE}$, where $c_3$ and $c_4$ are constants.

## 4 PLAYBOOK EXTENSION BY CLASS-INCREMENTAL LEARNING

When a new dataset is given, we can accommodate it efficiently by reusing existing skills while training new skills. For example, reusing the skill that reaches a specific position or object can be helpful because it is frequently performed for various manipulation tasks. Therefore, we aim to extend the playbook by adding new skills while reusing previously learned ones.

We assume that new datasets for the playbook extension are sequentially given. Due to memory limitations, we cannot store all previous data, so the dataset used for training is left with only a small amount. Then, we extend the playbook by adding new plays and primitives to accommodate both the remaining dataset and the given new dataset. As a result, the extended playbook owns diverse plays learned from different datasets. Finally, we focus on solving compounded problems, which are a mixture of old and new tasks, using an extended playbook.

### 4.1 CONTINUAL PLAY LEARNING FOR NEW DATASET

Since the play is a discrete variable, we can interpret a goal-conditioned RL problem using a playbook as a sequential play classification problem that selects play indices. Therefore, extending a playbook can be considered as a case of class-incremental learning problem. However, when performing class-incremental learning, we can face the problem of losing previously learned knowledge, which is called catastrophic forgetting. To extend the playbook while mitigating catastrophic forgetting, we apply the gradient boosting technique for a class-incremental learning method inspired by FOSTER (Wang et al., 2022). When the new dataset is given, FOSTER fixes previously trained models and adds new parameterized models to cover the new dataset. FOSTER aims to train additional models for the new dataset while maintaining the output of the original model for the remaining dataset. After training additional models, FOSTER compresses the entire model grown to the original model size through knowledge distillation. Since the state-action embedding model of the playbook selects plays, we perform class-incremental learning for the embedding model.

We extend the playbook in the following order. First, we freeze a pre-trained playbook consisting of a state-action embedding model and multiple plays and primitives. Next, we add a new parameterized embedding model and the fixed number of plays and primitives. Then, we define an extended embedding model that adds the output of the new and existing embedding models. Finally, the extended embedding model and the added plays and primitives are trained to minimize loss (3) for the new dataset. By training newly added plays and primitives, the expressive power of the playbook over the raw action space can be improved. Since we have the fixed original playbook, the extended embedding model can choose previously learned plays or train new plays when learning new tasks. After training the extended model, the embedding model is reduced to its original network size through knowledge distillation of FOSTER. In summary, when the playbook extension is completed, the size of the embedding model is maintained, and the number of plays and primitives increases. We refer to the above process of extending a playbook as a *continual play learning*.

## 5 SEQUENTIAL PLAY PLAN USING A PLAYBOOK

In this section, we explain the process of the play plan inference to reach a given goal state using a trained playbook. In general, all planning methods applicable in the discrete action space can be utilized for the playbook. In this paper, we propose two sampling-based methods, beam search and Monte Carlo tree search (MCTS) planners. The beam search planner is suitable for a playbook trained with a single dataset, and the MCTS planner addresses an extended playbook. These two planners have a rollout step and a selection step, as shown in Figure 3. First, in the rollout step, we imagine several future state-play sequences using the trajectory generation model $\Delta$. Next, in the

selection step, we select the best sequence that has reached the closest to the given goal state among the imagined sequences using the distance metric $\Psi$. We denote a planning set by $\{\Delta, \Psi\}$.

Before training a planning set, we transform the original dataset consisting of state-action sequences into state-play sequences using a trained playbook. With this dataset transformation, we gain two advantages. First, unlike raw actions, play is a discrete variable, so the complexity of state-play trajectories is reduced. Second, since action sequences are compressed into plays, the length of the state-play sequence is reduced. Therefore, as the number of inference steps required in the rollout step decreases, the computation cost and prediction error decrease. A more detailed explanation of the dataset transformation is presented in Appendix C.

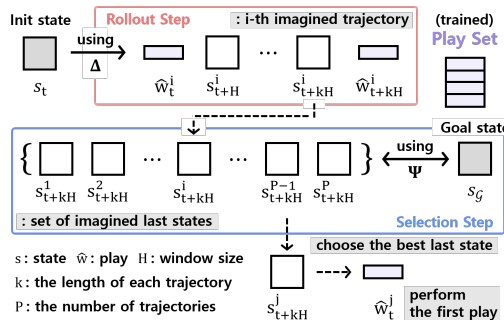

Figure 3: Play plan using planning sets.

## 5.1 PLAY PLAN GENERATION THROUGH BEAM SEARCH

We describe the beam search planner for inferring play sequences to reach a given goal state using a playbook, which is trained with a single dataset. In the rollout step of beam search, we use TT (Janner et al., 2021) as a trajectory generation model $\Delta$. The original TT learns the conditional probability distribution of state-action-reward sequences from the offline dataset. However, we only model state-play sequences because the task-agnostic dataset we use has no rewards. By utilizing TT, we obtain diverse and reasonable future state-play sequences conditioned on the given state.

In the selection step of beam search, we choose the best play sequence among the generated sequences. To this end, the playbook measures the dynamical distance in the state space between the last state of each plan and a given goal state. In goal-conditioned RL, the agent estimates Q-value, $Q(\boldsymbol{s}, \boldsymbol{s}_g, \boldsymbol{a})$, which indicates the discounted sum of rewards that can be obtained in the future if action $a$ is performed in the current state $s$ given the goal state $s_g$. If we use a sparse reward function for achieving goals, we can consider that a state with a higher Q-value reaches the goal in fewer time steps. Therefore, we use the Q-value estimated by the goal-conditioned offline RL algorithm with sparse rewards as a distance metric to measure the distance between two states.

We use IQL (Kostrikov et al., 2022), an offline RL algorithm, for a distance metric $\Psi$ by modifying IQL to fit the goal-conditioned RL setting, i.e., all states are concatenated with the goal state as the input for all parameterized models. To train goal-conditioned IQL, we need not only current states, plays, and the next states that can be sampled from the dataset but also goal states and rewards that are not given. Then, we sample goal states by setting the time step of the goal state as $t_G = t + \eta H$, which presents a time step after $\eta$ plays are performed. $H$ is a fixed window size, and $\eta \in \mathbb{N}$ is a random variable sampled along the geometric distribution. We use sparse rewards, which become $1$ if the goal state is reached within one play from the current state (i.e., $\eta = 1$) and $0$ if not.

Finally, through beam search with a planning set, the playbook can infer the play plan to reach a given goal state. The playbook converts the first play of the best plan into raw actions using primitives and performs the raw actions. Until the playbook achieves the goal state, the play plan process is repeated.

## 5.2 MONTE CARLO TREE SEARCH FOR MIXED-PLAY PLAN

We aim to find the mixed play plan to solve the compounded problems using an extended playbook. To this end, we propose an MCTS-based play planner that allows us to mix plays learned from different datasets freely. For performing MCTS, we train and retain the same planning sets as beam search, $\{\{\Delta^1, \Psi^1\}, \cdots, \{\Delta^P, \Psi^P\}\}$, for each of $P$ datasets. Note that the extended playbook must maintain all planning sets for each dataset, which is a limitation of the playbook extension.

We conduct a fixed number of tree searches, and each tree search generates one play plan. The process of tree search is as follows. First, MCTS repeats selecting one planning set $\{\Delta^i, \Psi^i\}$, which is trained from the dataset $\mathcal{D}^i$ among the owned sets until it reaches a leaf node or the maximum tree

depth. For all selections, MCTS infers a play and next state using $\Delta^i$, i.e., it is a rollout step. Next, when each tree search is completed, the value of the generated play plan is determined by measuring the distance between the last state and a given goal using $\Psi^i$, i.e., it is a selection step. By iteratively performing the above tree search, we can effectively generate mixed play plans. Finally, we choose the trajectory with the highest value among the inferred mixed plans. A more detailed explanation of the MCTS planner is presented in Appendix E.

## 6 EXPERIMENT

We conduct experiments to evaluate the playbook in complex and challenging environments. We focus on answering the following questions through experiments: 1) Can a finite number of plays cover a dataset consisting of task-agnostic demonstrations? In other words, can the playbook achieve better performance than existing methods? 2) Can the playbook extension successfully solve new tasks? 3) Can the extended playbook reuse previously learned plays when it solves new tasks? 4) How much do the IB objective and MCP structure affect the performance of the playbook?

### 6.1 EXPERIMENT SETUP

#### 6.1.1 BENCHMARK

We evaluate the playbook in simulated environments, which are selected based on the following characteristics: (1) the capability to perform diverse tasks sequentially within a single workspace, (2) the availability of a publicly accessible offline dataset, and (3) the possibility of obtaining a goal observation. Consequently, we utilize the following two environments in the experiments, as shown in Figure 4.

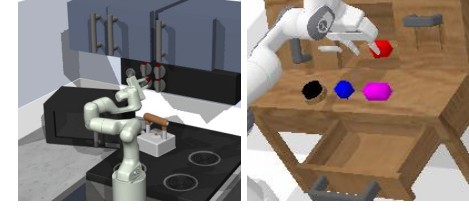

(a) Franka Kitchen      (b) CALVIN

Figure 4: Benchmarks used in experiments.

**Franka Kitchen** (Gupta et al., 2019) provides a kitchen workspace for manipulating various objects with a Franka robot, aiming to complete four predetermined sub-tasks consecutively. An observation is a 30-dimensional state representation, and an action is a 9-dimensional joint velocity vector for a robot arm.

**CALVIN** (Mees et al., 2022) is a benchmark for robotic manipulation tasks with a Franka robot on a desk. In this paper, we evaluate eight tasks related to a drawer, a slider, an LED, and a light bulb. We utilize an offline dataset in environment D of CALVIN and do not use task labels. An observation is a $3 \times 64 \times 64$-dimensional RGB image, and an action is a 7-dimensional robot action.

#### 6.1.2 BASELINES

**Offline RL algorithms**. We utilize CQL (Kumar et al., 2020), IQL (Kostrikov et al., 2022), and TT (Janner et al., 2021) with IQL as offline RL baselines for Franka Kitchen and CALVIN environments. We implement CQL and IQL and report their measured performance. On the other hand, we use the officially published code for TT. Particularly, TT+IQL is a method used for the play plan, allowing us to verify the performance difference between using raw actions and plays.

**Hierarchical policy learning methods.** We utilize Play-LMP (Lynch et al., 2019), RIL (Gupta et al., 2019), and TACO-RL (Rosete-Beas et al., 2022) as hierarchical policy learning baselines for the CALVIN environment. We select the above methods, which have the officially published code and allow goal-conditioned skill planning.

### 6.2 PERFORMANCE COMPARISON EXPERIMENT

#### 6.2.1 FRANKA KITCHEN RESULT

The playbook utilizes 32 plays and 16 primitives to cover the offline dataset of Franka Kitchen. Table 1 summarizes the performance results in Franka Kitchen. Since the Franka Kitchen benchmark provides sparse rewards, the cumulative reward signifies the number of completed sub-tasks. Note

| Dataset | CQL | IQL | TT+IQL | Playbook |
|---|---|---|---|---|
| Kitchen-Partial | $1.81 \pm 0.18$ (1.99) | $1.90 \pm 0.27$ (1.85) | $1.84 \pm 0.37$ | $\mathbf{2.32 \pm 0.42}$ |
| Kitchen-Mixed | $1.64 \pm 0.26$ (2.04) | $1.82 \pm 0.28$ (2.04) | $1.92 \pm 0.16$ | $\mathbf{2.50 \pm 0.20}$ |

Table 1: Performance results in Franka Kitchen. Each mean and standard deviation of the cumulative reward are calculated over 50 scenarios with three random seeds. Numbers in parentheses are the results reported in the cited papers.

| Number of Tasks | CQL | IQL | TT+IQL | Play-LMP | RIL | TACO-RL | Playbook |
|---|---|---|---|---|---|---|---|
| 1 | $0.143 \pm 0.035$ | $0.198 \pm 0.041$ | $0.402 \pm 0.119$ | 0.427 | 0.678 | 0.414 | $\mathbf{0.866 \pm 0.021}$ |
| 2 | $0.000 \pm 0.000$ | $0.000 \pm 0.000$ | $0.042 \pm 0.015$ | 0.039 | 0.221 | 0.165 | $\mathbf{0.508 \pm 0.037}$ |
| Average Length | 0.143 | 0.198 | 0.444 | 0.466 | 0.899 | 0.579 | $\mathbf{1.374}$ |

(a) Success rates for two sub-tasks chain problems

| Number of Tasks | CQL | IQL | TT+IQL | Play-LMP | RIL | TACO-RL | Playbook |
|---|---|---|---|---|---|---|---|
| 1 | $0.108 \pm 0.021$ | $0.135 \pm 0.023$ | $0.372 \pm 0.057$ | 0.400 | 0.701 | 0.213 | $\mathbf{0.901 \pm 0.011}$ |
| 2 | $0.000 \pm 0.000$ | $0.000 \pm 0.000$ | $0.082 \pm 0.019$ | 0.029 | 0.254 | 0.028 | $\mathbf{0.563 \pm 0.027}$ |
| 3 | $0.000 \pm 0.000$ | $0.000 \pm 0.000$ | $0.000 \pm 0.000$ | 0.000 | 0.013 | 0.000 | $\mathbf{0.214 \pm 0.021}$ |
| Average Length | 0.108 | 0.135 | 0.454 | 0.429 | 0.968 | 0.241 | $\mathbf{1.678}$ |

(b) Success rates for three sub-tasks chain problems

Table 2: Performance results for sub-task chains in CALVIN. Each mean and standard deviation of success rates are calculated over 1,000 scenarios with three random seeds. The average length indicates the average number of completed sub-tasks.

that the goal of Franka Kitchen is to perform four predetermined sub-tasks. On average, all offline RL algorithms only succeed in less than two sub-tasks, but the playbook completes more than two sub-tasks. In summary, the playbook outperforms offline RL baselines for all dataset types.

### 6.2.2 CALVIN RESULT

We conduct experiments in CALVIN requiring two or three sub-tasks to be performed sequentially to reach a given goal state, as depicted in Figure 5. It is challenging because we provide the agent with only one goal image, and the goal can only be achieved when all sub-tasks are completed. Therefore, the agent should consider both which tasks to perform and the order of tasks. For instance, if the agent is tasked with closing a drawer and placing a block inside it, the agent should position the block in the drawer before closing it.

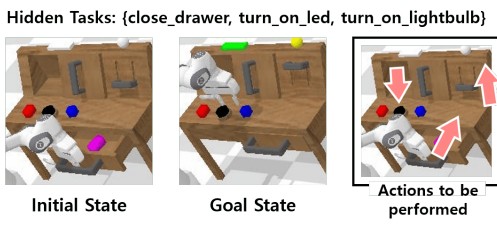

Figure 5: Example of three sub-task chain problem in CALVIN.

The playbook utilizes 64 plays and 32 primitives for the CALVIN benchmark. Table 2 shows the performance results in CALVIN, and we use the same trained model for each algorithm in both problems. First, offline RL methods show low performance, which means that using raw actions is disadvantageous for addressing long-horizon problems. On the other hand, hierarchical policy-based methods perform better, but on average, even one sub-task cannot be completed. In contrast, the playbook performs the best and succeeds in more than half of all sub-tasks, representing that using plays effectively solves long-horizon problems.

### 6.3 PLAYBOOK EXTENSION

In this experiment, we extend the trained playbook and evaluate it using complex image scenarios in the CALVIN benchmark. To accomplish this, we extract demonstrations of four predetermined sub-tasks (*close drawer*, *move slider left*, *turn on LED*, and *turn on lightbulb*) from the offline dataset of environment D of CALVIN. Consequently, we have five independent datasets: four task datasets,

| Model | Open Drawer | Move Slider Right | Turn off LED | Turn off Lightbulb | Close Drawer | Move Slider Left | Turn on LED | Turn on Lightbulb | Average |
|---|---|---|---|---|---|---|---|---|---|
| Init | $1.00 \pm 0.00$ | $0.96 \pm 0.04$ | $0.79 \pm 0.02$ | $0.95 \pm 0.02$ | $0.00 \pm 0.00$ | $0.05 \pm 0.02$ | $0.03 \pm 0.05$ | $0.04 \pm 0.07$ | 0.48 |
| Step 1 | $0.97 \pm 0.02$ | $0.96 \pm 0.04$ | $0.83 \pm 0.06$ | $0.89 \pm 0.02$ | $0.87 \pm 0.02$ | $0.05 \pm 0.02$ | $0.12 \pm 0.11$ | $0.05 \pm 0.02$ | 0.59 |
| Step 2 | $0.97 \pm 0.05$ | $0.96 \pm 0.04$ | $0.77 \pm 0.02$ | $0.92 \pm 0.04$ | $0.88 \pm 0.04$ | $0.76 \pm 0.04$ | $0.08 \pm 0.00$ | $0.07 \pm 0.02$ | 0.68 |
| Step 3 | $0.99 \pm 0.02$ | $0.96 \pm 0.04$ | $0.61 \pm 0.02$ | $0.91 \pm 0.02$ | $0.87 \pm 0.02$ | $0.76 \pm 0.07$ | $0.63 \pm 0.12$ | $0.04 \pm 0.04$ | 0.72 |
| Final | $0.99 \pm 0.02$ | $0.93 \pm 0.06$ | $0.60 \pm 0.16$ | $0.93 \pm 0.06$ | $0.77 \pm 0.02$ | $0.73 \pm 0.06$ | $0.56 \pm 0.18$ | $0.64 \pm 0.28$ | **0.77** |

Table 3: The success rate of the extended playbook for eight sub-tasks in CALVIN. We highlight the cell if the corresponding task dataset is not used for learning of each model. Each mean and standard deviation of success rate are averaged over 25 scenarios with three random seeds.

| Algorithm | Success Rate for Sequential Tasks | | Average Length |
|---|---|---|---|
| | 1 | 2 | |
| (Extended) Playbook | $0.659 \pm 0.058$ | $0.244 \pm 0.025$ | 0.903 |

Table 4: Performance results for sub-task chains of eight tasks in CALVIN using an extended playbook. Each mean and standard deviation of success rates are calculated over 100 scenarios with three random seeds. The average length indicates the average number of completed sub-tasks.

each involving the trajectories of a single task, and one base dataset containing data from other tasks. First, we train the initial playbook using the base dataset. And then, we progressively extend the playbook using four task datasets sequentially. We remove the previously used data for the playbook training, retaining only a ratio of $1\%$. The initial playbook has 64 plays and 32 primitives, and we add four plays and two primitives to enrich the expressive capability of the playbook when extending the playbook.

We incrementally extend a playbook by incorporating tasks in the order of *close drawer*, *move slider left*, *turn on LED*, and *turn on lightbulb*. Therefore, four continual play learning steps are required in total, so we obtain five trained models, including the initial playbook. For those five models, the success rates for eight sub-tasks are shown in Table 3. We find the best play plan via MCTS, as explained in Section 5.2. The results show that the extended playbook maintains the success rates of previously learned tasks and solves new tasks successfully through repeated continual play learning. Finally, the final model of the playbook successfully performs all eight sub-tasks.

Furthermore, we evaluate the extended playbook for compounded tasks, which are a mixture of old and new sub-tasks in CALVIN. Also, we use the MCTS planner to find the proper mixed play plan. We experiment with the final model, which has completed all continual play learning steps. Table 4 shows the performance results over 100 scenarios. The playbook achieves a success rate of $24.4\%$ for two sub-task chain problems, which indicates that the extended playbook can generate proper mixed play plans for compounded tasks.

### 6.4 ANALYSIS OF PLAY REUSE RATIO

To robustly and efficiently solve new tasks, it is important not only to learn new skills but also to reuse existing skills. Therefore, we analyze the selection rates of old and new plays when performing newly learned tasks using the extended playbook. We use the final model that has completed all continual learning steps in Section 6.3 and experiment in 25 episodes per task. The results of reuse ratios are shown in Figure 6. On average, when solving newly learned tasks, the agent chooses old plays at a rate of $28.8\%$, which means that old plays are useful in solving new tasks. Note that in the case of *turn on LED*, since *turn off LED* and *turn on LED* are performed by similar action sequences due to the desk structure of CALVIN, it is reasonable to use old plays more than new ones.

### 6.5 ABLATION STUDY

We conduct an ablation study for the playbook on CALVIN. Under the same experiment setting as three sub-task chains in Section 6.2.2, we identify the effect of the IB objective and MCP structure on the playbook performance. The first baseline algorithm, *playbook-$\alpha$*, maximizes only the first term of (4). In other words, the objective becomes the mutual information, excluding the information penalty term for action sequences. Next, the second algorithm, *playbook-$\beta$*, does not use the

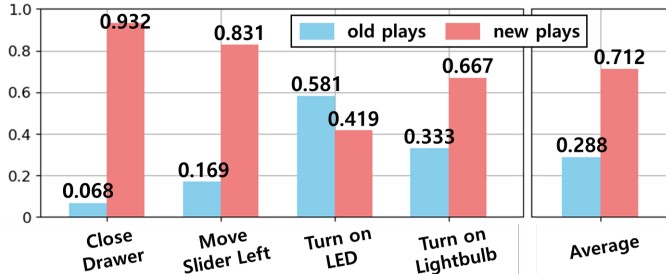

Figure 6: The play selection ratio for four newly learned tasks in CALVIN. Blue and red bars represent the selection ratio of old and new plays, respectively, when solving each task.

| Number of Tasks | playbook | playbook-$\alpha$ | playbook-$\beta$ | playbook-$\gamma$ | playbook-$\delta$ | playbook+BC |
|---|---|---|---|---|---|---|
| 1 | **0.901 ± 0.011** | 0.874 ± 0.012 | 0.882 ± 0.013 | 0.745 ± 0.174 | 0.878 ± 0.018 | 0.726 ± 0.018 |
| 2 | **0.563 ± 0.027** | 0.434 ± 0.006 | 0.477 ± 0.013 | 0.079 ± 0.013 | 0.533 ± 0.033 | 0.319 ± 0.026 |
| 3 | **0.214 ± 0.021** | 0.105 ± 0.008 | 0.154 ± 0.019 | 0.001 ± 0.001 | 0.152 ± 0.012 | 0.070 ± 0.050 |
| Average Length | **1.678** | 1.413 | 1.513 | 0.825 | 1.563 | 1.116 |

Table 5: Performance results of the ablation study for three sub-tasks chain problems. Each mean and standard deviation of success rates are calculated over 1,000 scenarios with three random seed The average length indicates the average number of completed sub-tasks.

IB objective, and action sequences are simply encoded through the action encoder. The third algorithm, *playbook-$\gamma$*, uses non-learnable one-hot vectors as plays to find out the effectiveness of MCP. In other words, the composite policy of the playbook-$\gamma$ becomes one primitive, not a combination of primitives. The fourth algorithm, *playbook-$\delta$*, forms the composite policy through a normalized linear combination of primitives rather than an exponential combination, i.e., it becomes a Gaussian mixture model. The last algorithm, *playbook+BC*, selects plays using a goal-conditioned behavioral cloning model instead of the planning set. As a result, the original playbook shows the best performance of 1.678, as shown in Table 5.

## 7 LIMITATIONS

**Remaining all planning sets for mixed play plan.** As mentioned in Section 5.2, we must preserve planning sets for all datasets in order to generate mixed play plans using an extended playbook. Since planning sets are composed of offline RL algorithms, this limitation can be solved through continual RL for planning sets, but it is still a challenging problem. This limitation can be addressed in the future work for the playbook.

**Trade-off between the number of plays and planning efficiency.** The playbook has a trade-off between the number of plays and the efficiency of the play plan. As the number of plays increases, the ability of the playbook to cover multi-modal actions improves, but as the complexity of the play plan increases, the required computational cost increases and the efficiency decreases. Conversely, as the number of plays decreases, the play plan becomes simple, but the expressive power of the playbook decreases. Therefore, the playbook must explore the appropriate number of plays experimentally.

## 8 CONCLUSION

In this paper, we propose a novel scalable offline discrete skill discovery algorithm, a *playbook*, for long-horizon decision-making problems. The playbook provides a straightforward way to expanding the skill set by utilizing discrete skills and the extensible structure. Furthermore, the playbook effectively expresses multi-modal behavior distributions included in the dataset with only a finite number of skills. Experimentally, we confirm that the playbook with a discrete skill set performs better than existing baselines, which utilize the continuous skill space. In addition, we verify that the extended playbook not only successfully solves tasks included in new datasets but also carries out compounded tasks, which are a mixture of old and new tasks.

REPRODUCIBILITY STATEMENT

We reported hyperparameter settings for the playbook training and inference in Appendix D.1, F.2, G.2, and H.2. We also described the experiment settings in detail in the main paper and Appendix F, G, and H. The source code for reproducing our reported results can be found in the supplementary material.

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

## A  Multiplicative Compositional Policy in a Playbook

A playbook has $M$ parameterized primitives $\{\pi_{\phi_1}, \cdots, \pi_{\phi_M}\}$, and each primitive presents an independent probability distribution over a raw action space, $p_{\pi_{(\cdot)}}(\boldsymbol{a}|\boldsymbol{s})$, taking a state $\boldsymbol{s}$ as an input. We model each primitive as a Gaussian distribution with mean $\mu_m(\boldsymbol{s})$ and diagonal covariance matrix $\Sigma_m(\boldsymbol{s})$. According to the formula of the composite policy in Peng et al. (2019), the output of the composite policy $\hat{\pi}$ is derived as a Gaussian distribution with the following mean and covariance:

$$\mu^j(\boldsymbol{s}, \hat{w}) = \frac{1}{\sum\limits_{k=1}^{M} \frac{\hat{w}^k}{\sigma_k^j(\boldsymbol{s})}} \sum_{m=1}^{M} \frac{\hat{w}^m}{\sigma_m^j(\boldsymbol{s})} \mu_m^j(\boldsymbol{s}), \quad \sigma^j(\boldsymbol{s}, \hat{w}) = \left( \sum_{m=1}^{M} \frac{\hat{w}^m}{\sigma_m^j(\boldsymbol{s})} \right)^{-1}, \tag{6}$$

where $\hat{w}^m$ is the $m$-th element of the weight vector $\hat{w}$, $\mu^j(\boldsymbol{s}, \hat{w})$ and $\sigma^j(\boldsymbol{s}, \hat{w})$ are the $j$-th element of mean and variance of the composite distribution $\hat{\pi}(\cdot|\boldsymbol{s}, \hat{w})$, and $\mu_m^j(\boldsymbol{s})$ and $\sigma_m^j(\boldsymbol{s})$ are the $j$-th element of mean and variance of primitive $\pi_{\phi_m}(\cdot|\boldsymbol{s})$. Using the above distribution, we can calculate the probability $\hat{\pi}(\boldsymbol{a}|\boldsymbol{s}, \hat{w})$ of an action $\boldsymbol{a}$ to be chosen for a given state-play pair $(\boldsymbol{s}, \hat{w})$.

## B  Lower Bound for the Information Bottleneck Objective

In Section 3.3, we propose the following IB objective (4) for training the action encoder parameterized by $\psi$ and the latent mapping model parameterized by $\xi$:

$$\text{maximize} \ \ \mathbb{E}_t \left[ \mathcal{I}(Z^{s_{t+H}}; Z^{a_{t:t+H}}|S_t) - \beta \mathcal{I}(Z^{a_{t:t+H}}; A_{t:t+H}) \right],$$

where $S_t$, $A_{t:t+H}$, $Z^{s_t}$, and $Z^{a_{t:t+H}}$ are random variables corresponding to $\boldsymbol{s}_t$, $\boldsymbol{a}_{t:t+H}$, $\mathbf{z}^{s_t}$, and $\mathbf{z}^{a_{t:t+H}}$, respectively, and $\beta$ is a constant. Also, $\mathbf{z}^s$ and $\mathbf{z}^a$ are a state latent and an action latent that encode a state and an action sequence, respectively. First, we use the action encoder $\psi$ to embed an action sequence into an action latent $\mathbf{z}^{a_{t:t+H}}$. Next, we define the latent mapping model $\xi$ as $p_\xi(\mathbf{z}^{s_{t+H}}|\boldsymbol{s}_t, \mathbf{z}^{a_{t:t+H}})$, which predicts the distribution of the future state latent as Gaussian using the current state and action latent as an input. In this section, we derive its lower bound (5) to maximize the above objective.

The first term, $\mathcal{I}(Z^{s_{t+H}}; Z^{a_{t:t+H}}|S_t)$, means that given the current state $\boldsymbol{s}_t$, an action latent $\mathbf{z}^{a_{t:t+H}}$ is informative about a state latent $\mathbf{z}^{s_{t+H}}$. We derive the following approximated lower bound of the mutual information using a variational approximation of $p_\xi(\mathbf{z}^{s_{t+H}}|\boldsymbol{s}_t, \mathbf{z}^{a_{t:t+H}})$.

$$\mathbb{E}_t \left[ \mathcal{I}(Z^{s_{t+H}}; Z^{a_{t:t+H}}|S_t) \right] = \mathop{\mathbb{E}}_{\tau \sim \mathcal{D}, t, z^{s_{t+H}} \sim p_\omega(\cdot|\tau), z^{a_{t:t+H}} \sim p_\psi(\cdot|\tau)} \left[ \log \frac{p_\omega(\mathbf{z}^{s_{t+H}}|\boldsymbol{s}_t, \mathbf{z}^{a_{t:t+H}})}{p_\omega(\mathbf{z}^{s_{t+H}}|\boldsymbol{s}_t)} \right]$$

$$\geq \mathbb{E}_{\tau, t, z^{s_{t+H}}, z^{a_{t:t+H}}} \left[ \log p_\xi(\mathbf{z}^{s_{t+H}}|\boldsymbol{s}_t, \mathbf{z}^{a_{t:t+H}}) - \log p_\omega(\mathbf{z}^{s_{t+H}}|\boldsymbol{s}_t) \right] \tag{7}$$

$$\approx \mathbb{E}_{\tau, t, z^{s_{t+H}}, z^{a_{t:t+H}}} \left[ \log p_\xi(\mathbf{z}^{s_{t+H}}|\boldsymbol{s}_t, \mathbf{z}^{a_{t:t+H}}) - \log \mathbb{E}_{z^a} \left[ p_\xi(\mathbf{z}^{s_{t+H}}|\boldsymbol{s}_t, \mathbf{z}^a) \right] \right].$$

Next, minimizing the second term, $\mathcal{I}(Z^{a_{t:t+H}}; A_{t:t+H})$, means that the action latent $\mathbf{z}^{a_{t:t+H}}$ is penalized for preserving information about the action sequence $\boldsymbol{a}_{t:t+H}$. This term has the following upper bound:

$$\mathbb{E}_t \left[ \mathcal{I}(Z^{a_{t:t+H}}; A_{t:t+H}) \right] = \mathop{\mathbb{E}}_{\tau \sim \mathcal{D}, t, z^{a_{t:t+H}} \sim p_\psi(\cdot|\tau)} \left[ \log \frac{p_\psi(\mathbf{z}^{a_{t:t+H}}|\boldsymbol{a}_{t:t+H})}{p_\psi(\mathbf{z}^{a_{t:t+H}})} \right]$$

$$\leq \mathbb{E}_{\tau, t, z^{a_{t:t+H}}} \left[ D_{KL}\left( p_\psi(Z^{a_{t:t+H}}|\boldsymbol{a}_{t:t+H}) \parallel q(Z^{a_{t:t+H}}) \right) \right], \tag{8}$$

where $q(z^a)$ is a normal distribution, $\mathcal{N}(0, I)$ used as a variational approximation of the prior distribution $p_\psi(\mathbf{z}^a)$. Finally, we obtain the following loss for the action encoder and latent mapping model, i.e., $\psi$ and $\xi$:

$$\mathcal{L}_{\{\psi, \xi\}} = - \mathop{\mathbb{E}}_{\tau, t, z^{s_{t+H}}, z^{a_{t:t+H}}} \left[ \log p_\xi(\mathbf{z}^{s_{t+H}}|\boldsymbol{s}_t, \mathbf{z}^{a_{t:t+H}}) \right.$$

$$\left. - \log \mathbb{E}_{z^a} \left[ p_\xi(\mathbf{z}^{s_{t+H}}|\boldsymbol{s}_t, \mathbf{z}^a) \right] - \beta D_{KL}\left( p_\psi(Z^{a_{t:t+H}}|\boldsymbol{a}_{t:t+H}) \parallel q(Z^{a_{t:t+H}}) \right) \right].$$

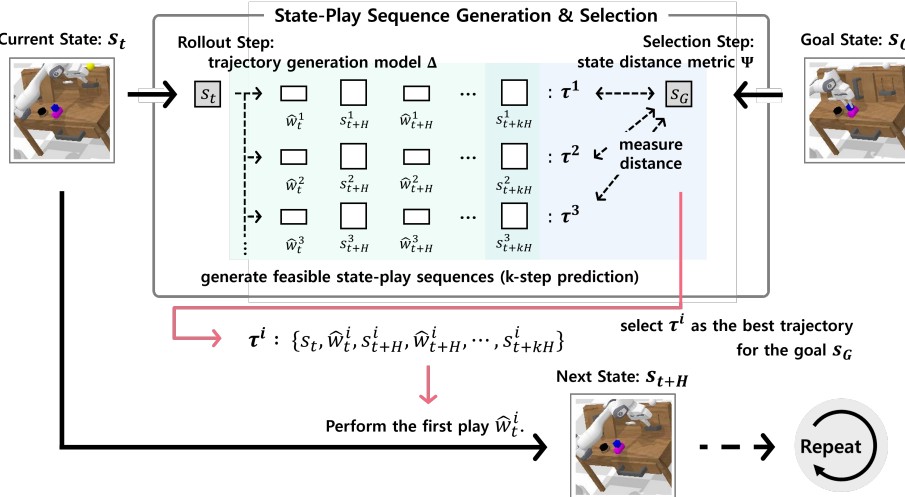

Figure 7: A playbook converts original demonstrations into state-play sequences using a pre-trained playbook for training play planning sets.

Figure 8: Overview of a play plan through beam search. The playbook generates various state-play sequences from the current state through repeated rollout steps and selects the best sequence closest to a given goal state through a selection step.

## C    OFFLINE TRAJECTORY PROCESSING USING A PRE-TRAINED PLAYBOOK

Before training the planning sets, we convert pre-collected demonstrations into state-play trajectories using a pre-trained playbook, as depicted in Figure 7. The playbook selects a play $\hat{w}_t \in \mathcal{B}$ by utilizing the current state $s_t$ and window size actions $a_{t:t+H}$ as an input. Therefore, original state-action sequences of the unstructured dataset are converted into state-play sequences. The converted sequence has fewer steps than the original one, but the play maintains information about the original action sequence, making it easier to predict distant future states than a one-step dynamics model. In other words, the playbook can reduce accumulated prediction errors caused by repeated predictions for state transitions.

## D    PLAY SELECTION PROCESS THROUGH BEAM SEARCH

This section deals with the state-play sequence inference process using trained planning sets. The planning process follows beam search proposed by Janner et al. (2021). Figure 8 shows the process of selecting the best play index using a pre-trained playbook to reach a goal state. First, the playbook uses the trajectory generation model $\Delta$ to generate diverse and reasonable state-play sequences from the current state. Next, the playbook uses the state distance metric $\Psi$ to select the closest trajectory to a given goal state. The first play of the selected sequence is converted into raw actions using a low-level policy and is performed in the current state. This process is repeated until the goal state is reached.

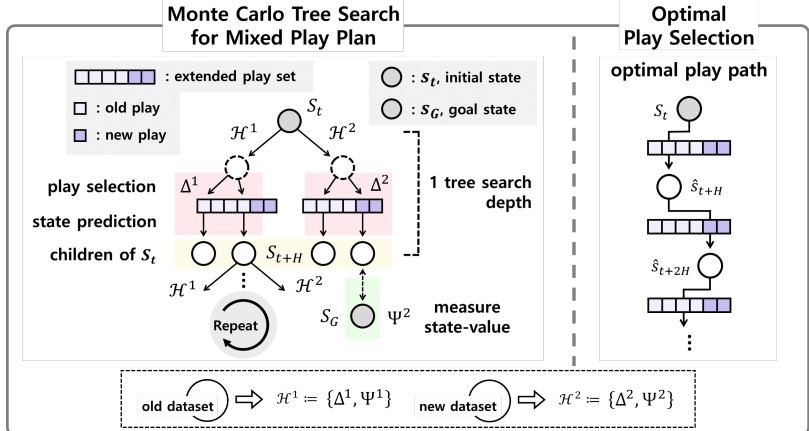

Figure 9: Overview of MCTS for generating a mixed play plan. (Left) The playbook performs a tree search by sequentially selecting a planning set $\{\Delta^i, \Psi^i\}$. In each step in the tree search, we reach the next node by inferring a play and next state using the selected $\Delta^i$. If we reach a leaf node, the value of the node is measured using $\Psi^i$. (Right) After the tree search is finished, the playbook sequentially selects the play index with the highest node value to infer the optimal play plan.

### D.1 Hyperparameter Setting for a Play Plan

In experiments 6.2.1, 6.2.2, 6.4, and 6.5, we perform beam search for a play plan to infer the optimal play sequence with the same hyperparameter setting. First, we generated $64$ independent state-play trajectories and performed eight rollout steps with a window size of $10$, i.e., the future state after $80$ time steps is predicted in each sequence.

## E  Mixed-Play Plan through Monte Carlo Tree Search

### E.1  Tree Search Phase

We generates play sequences with an extended playbook by mixing plays learned from multiple datasets using the MCTS planner, as shown in Figure 9. MCTS is performed through iterative tree search processes. In each tree search process, the planner sequentially chooses a planning set $\{\Delta^i, \Psi^i\}$ among all planning sets at the current node. Next, the planner determines the next node by deciding a play index to be performed and predicting the next state using $\Delta^i$. Then, the depth of the tree search increases by one. If the planner reaches a leaf node, the value of the leaf node is measured by $\Psi^i$ using a given goal state. Each tree search process is performed until a leaf node is reached or the maximum depth of the tree is achieved.

### E.2  Optimal Play Plan Generation Phase

After MCTS is finished, the planner finds the best play sequence within the searched tree based on the stored node values. By starting from the root node, the planner selects the play index with the highest node value among all plays owned by the current node. Until a leaf node is reached, the planner generates the best play sequence by sequentially selecting the play indices. In practice, we execute only the first play in the inferred play plan. Then, we obtain the next state by executing raw actions in the environment and perform MCTS again until the goal state is achieved.

## F  Franka Kitchen Experiment

### F.1  Environment Setting

In the Franka Kitchen environment (Fu et al., 2020), a Franka arm robot achieves a given goal state by manipulating various objects in a virtual kitchen. The kitchen has available objects such as a

kettle, a light switch, a microwave, an opening cabinet, and a sliding cabinet. Experiments are conducted in two environments: *kitchen-partial-v0* and *kitchen-mixed-v0*[1]. In both environments, the goal state is fixed with the following four target sub-tasks completed: *open the micro wave*, *move the kettle*, *flip the light switch*, and *slide open the cabinet door*.

In experiments, we use *partial* and *mixed* offline datasets, which are challenging datasets of Franka Kitchen. In both datasets, all offline trajectories include demonstrations that carry out sub-tasks that are not part of the target sub-tasks. There are trajectories that perform all target sub-tasks in the partial-type dataset but not in the mixed-type dataset.

## F.2    IMPLEMENTATION DETAILS FOR BASELINES

We use CQL (Kumar et al., 2020), IQL (Kostrikov et al., 2022), and TT+IQL (Janner et al., 2021) as baselines in Franka Kitchen. First, we implemented CQL and IQL algorithms. For CQL and IQL, since the performance results for Franka Kitchen are reported in each reference paper, we list both the performance we measured and the performance reported by the authors in Table 1. On the other hand, we modified the official code of TT[2] to fit our experimental setting. Since TT has no reported performance for Franka Kitchen, we only list the performance we measured. In particular, TT+IQL is an important baseline, which is a method used in plan plan for the playbook, allowing us to confirm the performance difference between using raw actions and plays.

## F.3    HYPERPARAMETER SETTING FOR A PLAYBOOK

In Franka Kitchen, a playbook is trained with the following hyperparameter settings. First, the hyperparameter setting used for training the playbook is shown in Table 6.

| Hyperparameter | Value | Hyperparameter | Value |
|---|---|---|---|
| window size ($H$) | 10 | learning rate | 3e-4 |
| batch size | 128 | training steps | 3e5 |
| dimension of $z^d$ | 32 | the number of plays ($N$) | 32 |
| dimension of $z^i$ | 16 | the number of primitives ($M$) | 16 |

Table 6: Hyperparameter setting for training the playbook.

Next, the hyperparameter setting for training a trajectory generation model is shown in Table 7. Other hyperparameters are set to the default value of the official code of TT.

| Hyperparameter | Value | Hyperparameter | Value |
|---|---|---|---|
| the number of quantizations | 100 | the number of attention layers | 4 |
| training steps | 5e5 | the number of attetion heads | 4 |

Table 7: Hyperparameter setting for training a trajectory generation model.

Lastly, the hyperparameter setting used for training a state distance metric is shown in Table 8. We used an IQL code we implemented.

| Hyperparameter | Value | Hyperparameter | Value |
|---|---|---|---|
| window size ($H$) | 10 | learning rate | 3e-4 |
| batch size | 128 | training steps | 2e5 |
| temperature | 0.7 | tau | 1e-3 |
| expectile | 0.5 | discount factor | 0.99 |
| alpha | 2.0 | | |

Table 8: Hyperparameter setting for training a state distance metric of high-level models.

---

[1] https://github.com/Farama-Foundation/D4RL

[2] https://github.com/jannerm/trajectory-transformer

# G  CALVIN EXPERIMENT

## G.1  ENVIRONMENT SETTING

In the CALVIN environment (Mees et al., 2022), the robot agent achieves a given goal state by manipulating various objects on a multifunctional desk. There is a drawer, a slider, an LED, a lightbulb, and three blocks on the desk. CALVIN provides offline demonstrations that were collected by humans who controlled a robot arm via teleoperation[3]. Each demonstration is a long trajectory in which diverse sub-tasks are sequentially performed in random order. Note that the dataset provides task labels, but we do not use them for playbook learning.

## G.2  IMPLEMENTATION DETAILS FOR BASELINES

We use offline RL algorithms (CQL (Kumar et al., 2020), IQL (Kostrikov et al., 2022), and TT+IQL (Janner et al., 2021)) and hierarchical policy learning algorithms (Play-LMP (Lynch et al., 2019), RIL (Gupta et al., 2019), and TACO-RL (Rosete-Beas et al., 2022)) as baselines CALVIN. For CQL, IQL, and TT+IQL, we measured the performance using the code we implemented. On the other hand, for Play-LMP, RIL, and TACO-RL, we utilized saved checkpoint models[4]. For fair comparison with a playbook, we used the same model for sub-task chain problems for all baselines.

## G.3  HYPERPARAMETER SETTING FOR A PLAYBOOK

In CALVIN, a playbook is trained with the following hyperparameter settings. First, the hyperparameter setting used for training the playbook is shown in Table 9.

| Hyperparameter | Value | Hyperparameter | Value |
|---|---|---|---|
| window size ($H$) | 10 | learning rate | 3e-4 |
| batch size | 128 | training steps | 3e5 |
| dimension of $z^d$ | 64 | the number of plays ($N$) | 64 |
| dimension of $z^i$ | 32 | the number of primitives ($M$) | 32 |

Table 9: Hyperparameter setting for training the playbook.

Next, the hyperparameter setting for training a trajectory generation model is shown in Table 10. We used the official code of TT for the trajectory generation model, and other hyperparameters are set to the default value of the code.

| Hyperparameter | Value | Hyperparameter | Value |
|---|---|---|---|
| the number of quantizations | 100 | the number of attention layers | 4 |
| training steps | 1e6 | the number of attention heads | 4 |

Table 10: Hyperparameter setting for training a trajectory generation model.

Lastly, the hyperparameter setting used for training a state distance metric is shown in Table 11. We used an IQL code we implemented.

| Hyperparameter | Value | Hyperparameter | Value |
|---|---|---|---|
| window size ($H$) | 10 | learning rate | 3e-4 |
| batch size | 128 | training steps | 1e6 |
| temperature | 10.0 | tau | 1e-3 |
| expectile | 0.9 | discount factor | 0.99 |
| alpha | 2.0 | geometric probability ($p_{\mathcal{G}}$) | 0.10 |

Table 11: Hyperparameter setting for training a state distance metric.

---

[3]https://github.com/mees/calvin

[4]https://github.com/ErickRosete/tacorl

### G.4 EXECUTION RESULTS USING PLAYBOOK

To help understand experiments in the CALVIN environment, we visualize successful examples of execution results using a playbook in two and three sub-task chain problems in Figure 10.

## H PLAYBOOK EXTENSION EXPERIMENT IN CALVIN

### H.1 SIZE OF BASE AND TASK DATASETS

Table 12 shows the size of the base dataset and task datasets used in Section 6.3. The base dataset is the largest because it contains demonstrations for all sub-tasks except for four predetermined sub-tasks: *close drawer*, *move slider left*, *turn on LED*, and *turn on lightbulb*. We performed continual play learning using *close drawer*, *move slider left*, *turn on LED*, and *turn on lightbulb* datasets in order for the playbook extension. After each continual play learning is finished, we leave only 1% of data for each dataset we used for the subsequent continual play learning.

| Dataset | Num. of Time Steps |
|---|---|
| Base Dataset | 400,633 |
| Close-Drawer Dataset | 56,088 |
| Move-Slider-Left Dataset | 66,719 |
| Turn-on-LED Dataset | 36,890 |
| Turn-on-Lightbulb Dataset | 36,660 |

Table 12: The size of the base dataset and task datasets used for continual play learning.

### H.2 HYPERPARAMETER SETTING FOR PLAYBOOK EXTENSION

A playbook is expanded via a total of four continual play learning steps in Section 6.3. The hyperparameter setting for training the playbook of each continual play learning step is the same, which is shown in Table 13. Other hyperparameter settings for training planning sets are the same as Appendix G.3.

| Hyperparameter | Value | Hyperparameter | Value |
|---|---|---|---|
| window size ($H$) | 10 | learning rate | 3e-4 |
| batch size | 128 | the number of plays ($N$) | 64 |
| dimension of $z^d$ | 64 | the number of primitives ($M$) | 32 |
| dimension of $z^i$ | 32 | the number of additional plays | 4 |
| training steps: initial phase | 3e5 | the number of additional primitives | 2 |
| training steps: extension phase | 2e5 | remaining ratio for old data | 0.01 |
| training steps: distillation phase | 1e5 | | |

Table 13: Hyperparameter setting for training the playbook for a playbook extension.

## I PLAYBOOK WITH GOAL-CONDITIONED BEHAVIORAL CLONING

This section details the *playbook-BC* algorithm used in the ablation study in Section 6.5. For the CALVIN benchmark, because the goal-conditioned behavioral cloning (GCBC) model trained on the low-level action space shows low performance, we train the GCBC model on the high-level action space, i.e., play set, to measure its performance. Like the training of the planning set in Section 5, the GCBC model is trained using a converted dataset with a pre-trained playbook. GCBC infers plays to be performed with the current and goal states as inputs. For training GCBC, the current states and plays are sampled directly from the converted dataset, and the goal states are sampled among future states through the same process in Section 5.1.

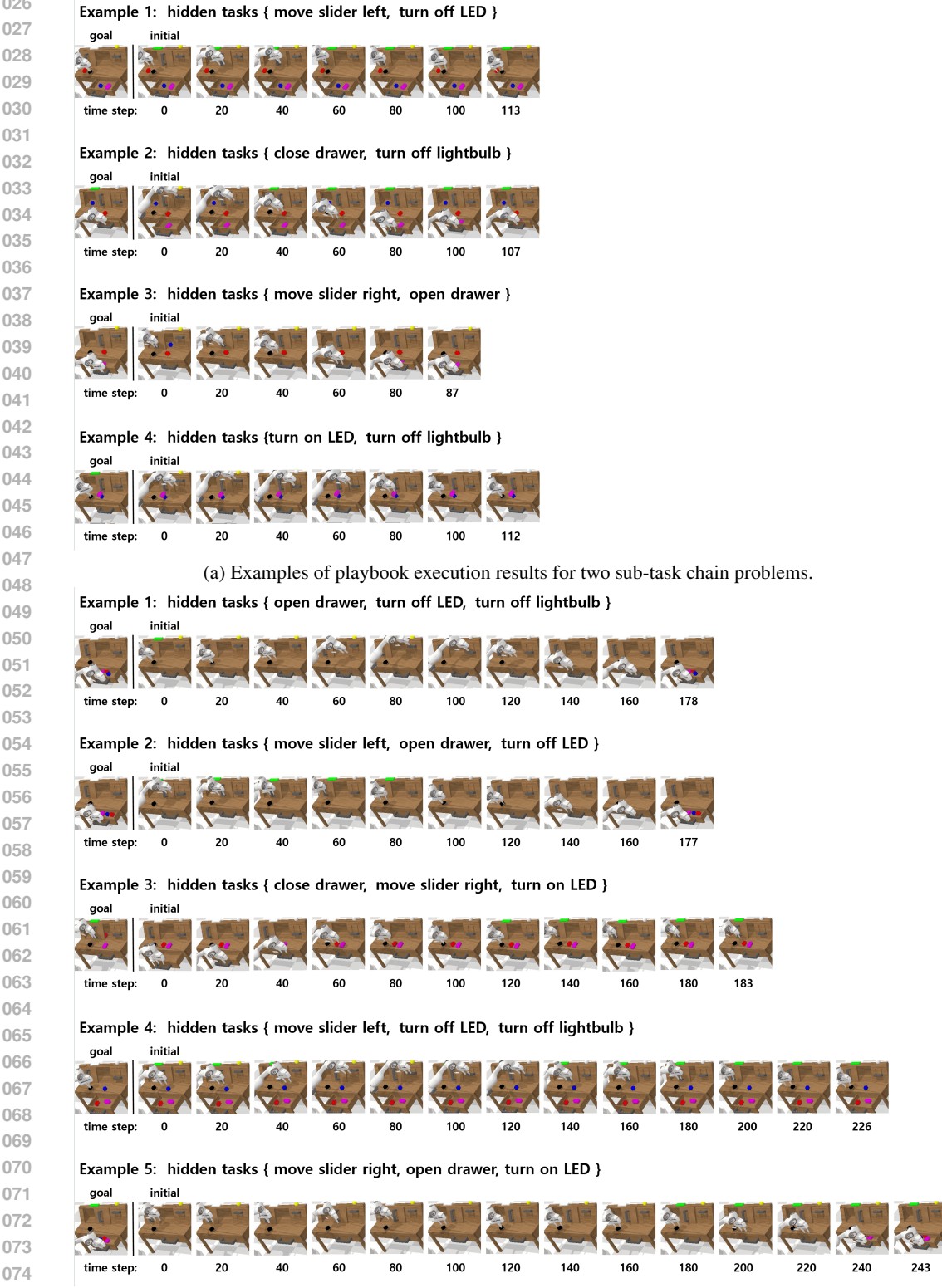

(a) Examples of playbook execution results for two sub-task chain problems.

(b) Examples of playbook execution results for three sub-task chain problems.

Figure 10: Execution results in the CALVIN environment.

| The Number | Success Rate for Sequential Tasks | | | Average |
| of Plays | 1 | 2 | 3 | Length |
|---|---|---|---|---|
| 32 | $0.855 \pm 0.035$ | $0.482 \pm 0.025$ | $0.163 \pm 0.030$ | 1.500 |
| 64 | $0.901 \pm 0.011$ | $0.563 \pm 0.027$ | $0.214 \pm 0.021$ | **1.678** |
| 128 | $0.867 \pm 0.029$ | $0.567 \pm 0.026$ | $0.213 \pm 0.039$ | 1.647 |

Table 14: Performance results for three sub-task chains in CALVIN using playbooks with different number of plays. Each mean and standard deviation of success rates are calculated over 1000 scenarios with three random seeds. The average length indicates the average number of completed sub-tasks.

| The Number | Success Rate for Sequential Tasks | | | Average |
| of Primitives | 1 | 2 | 3 | Length |
|---|---|---|---|---|
| 16 | $0.864 \pm 0.051$ | $0.514 \pm 0.022$ | $0.181 \pm 0.027$ | 1.559 |
| 32 | $0.901 \pm 0.011$ | $0.563 \pm 0.027$ | $0.214 \pm 0.021$ | **1.678** |
| 64 | $0.876 \pm 0.028$ | $0.569 \pm 0.010$ | $0.220 \pm 0.020$ | 1.665 |

Table 15: Performance results for three sub-task chains in CALVIN using playbooks with different number of primitives. Each mean and standard deviation of success rates are calculated over 1000 scenarios with three random seeds. The average length indicates the average number of completed sub-tasks.

| Window | Success Rate for Sequential Tasks | | | Average |
| Size | 1 | 2 | 3 | Length |
|---|---|---|---|---|
| 5 | $0.830 \pm 0.033$ | $0.467 \pm 0.043$ | $0.180 \pm 0.015$ | 1.477 |
| 10 | $0.901 \pm 0.011$ | $0.563 \pm 0.027$ | $0.214 \pm 0.021$ | **1.678** |
| 20 | $0.870 \pm 0.025$ | $0.468 \pm 0.051$ | $0.125 \pm 0.031$ | 1.463 |

Table 16: Performance results for three sub-task chains in CALVIN using playbooks with different window sizes. Each mean and standard deviation of success rates are calculated over 1000 scenarios with three random seeds. The average length indicates the average number of completed sub-tasks.

| Horizon | Success Rate for Sequential Tasks | | | Average |
| | 1 | 2 | 3 | Length |
|---|---|---|---|---|
| 1 | $0.826 \pm 0.015$ | $0.369 \pm 0.025$ | $0.094 \pm 0.042$ | 1.289 |
| 3 | $0.897 \pm 0.031$ | $0.555 \pm 0.033$ | $0.168 \pm 0.035$ | 1.620 |
| 5 | $0.891 \pm 0.011$ | $0.575 \pm 0.033$ | $0.192 \pm 0.009$ | 1.658 |
| 8 | $0.901 \pm 0.011$ | $0.563 \pm 0.027$ | $0.213 \pm 0.039$ | **1.678** |

Table 17: Performance results for three sub-task chains in CALVIN using playbooks with beam search horizons. Each mean and standard deviation of success rates are calculated over 1000 scenarios with three random seeds. The average length indicates the average number of completed sub-tasks.

## J    ADDITIONAL ABLATION STUDIES

In this section, we conduct additional ablation studies to confirm the performance change caused by the adjustment of hyperparameters of the playbook. These experiments address three sub-task chain problems under the same experimental setting as Section 6.2.2. The original playbook uses 64 plays and 32 primitives and has a window size of 10 when training. Also, when performing beam search, an agent infers the state after eight plays have been performed. Therefore, we check the effect of the number of plays, the number of primitives, the window size, and the inference horizon of beam search on the performance of the playbook.

Tables 14 and 15 present the performance results according to the number of plays and primitives. As a result, the playbook shows the best performance when using 64 plays or 32 primitives. The

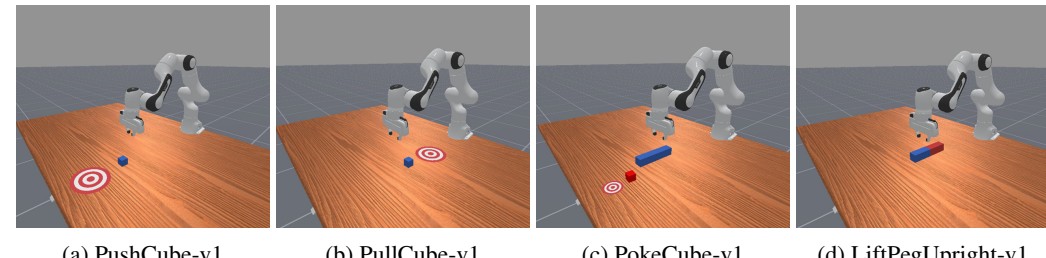

| (a) PushCube-v1 | (b) PullCube-v1 | (c) PokeCube-v1 | (d) LiftPegUpright-v1 |

Figure 11: ManiSkill3 tasks used in experiments.

| Task | BC | CQL | IQL | Playbook+BC |
|------|-----|-----|-----|-------------|
| PushCube-v1 | $0.67 \pm 0.06$ | $0.57 \pm 0.13$ | $0.67 \pm 0.04$ | $\mathbf{0.76 \pm 0.04}$ |
| PullCube-v1 | $0.45 \pm 0.11$ | $0.36 \pm 0.11$ | $0.43 \pm 0.15$ | $\mathbf{0.58 \pm 0.04}$ |
| PokeCube-v1 | $0.56 \pm 0.13$ | $0.41 \pm 0.09$ | $0.58 \pm 0.09$ | $\mathbf{0.66 \pm 0.10}$ |
| LiftPegUpright-v1 | $0.30 \pm 0.12$ | $0.16 \pm 0.03$ | $0.19 \pm 0.04$ | $\mathbf{0.46 \pm 0.08}$ |
| Average | $0.50 \pm 0.16$ | $0.38 \pm 0.17$ | $0.47 \pm 0.21$ | $\mathbf{0.62 \pm 0.13}$ |

Table 18: Performance results in the ManiSkill3 benchmark. Each mean and standard deviation of the success rate are calculated over 100 episodes with three random seeds.

playbook with 32 plays or 16 primitives records low performance, indicating that the number of components is insufficient to express multi-modal behaviors. On the other hand, the playbook with 128 plays or 64 primitives shows slightly lower results than the highest performance. This means that if the number of plays and primitives is larger than necessary, the complexity of planning increases, and then performance results can deteriorate.

Table 16 presents the performance results of the playbook with different window sizes. Depending on the window size, the playbook has the following trade-off. If the window size is small, raw action inference for each play becomes accurate, but more rollout steps are required when play planning. On the other hand, when the window size is large, the accuracy of raw action inference for each play decreases, but only a small number of rollout steps can predict a distant future state. We experimentally confirm that the window size of 10 shows the best performance.

Finally, Table 17 presents the performance results of the playbook with different planning horizons. The results show that the performance of the playbook improves as the planning horizon increases, which implies that an agent can successfully predict future states through beam search.

## K  ADDITIONAL BENCHMARK FOR PLAYBOOK EVALUATION

In this section, we experiment with playbook learning using an offline dataset consisting of several single-task demonstrations. To this end, we utilize the ManiSkill3 (Tao et al., 2024) benchmark, a SAPIEN (Xiang et al., 2020)-based simulated environment and solve the following robot manipulation tasks: *PushCube-v1*, *PullCube-v1*, *PokeCube-v1*, and *LiftPegUpright-v1*, as shown in Figure 11. The differences between the experiments in ManiSkill3 and CALVIN are as follows. First, in ManiSkill3, rather than sequentially performing multiple tasks in a single workspace, only one task is performed in one workspace. Thus, the agent can perform proper tasks without goal states. Second, since there is no public dataset for the above tasks, we collect the offline dataset for training.

**Offline data collection.** The ManiSkill3 benchmark provides a motion planning-based framework for collecting task demonstrations, but these collected trajectories make it challenging to train the agent robustly due to a lack of diversity. Thus, for diverse data collection, we train a reinforcement learning agent for each task and then obtain successful demonstrations from various checkpoints. We use SAC Haarnoja et al. (2018) as a reinforcement learning algorithm and train the agent using the dense reward function provided by ManiSkill3. An observation is a state representation that includes the position of the end effector, objects, and goal region, and an action represents the pose of the end effector of the robot. Finally, we collected 10,000 demonstrations for each task. Since

each task has a different dimension of the observation, we apply zero padding to each state so that all states have the same dimensions.

**Analyzing the experiment results.** Since we solve tasks without goals, we evaluate the playbook by training the BC model for the play set without utilizing a planning set. The experimental results of ManiSkill3 can be found in Table 18. We measure the average success rate for 100 episodes with three random seeds for each task. First, BC, CQL, and IQL methods show success rates of 0.50, 0.38, and 0.47 on average for all tasks, respectively. Meanwhile, the playbook+BC records the best success rate of 0.62. Compared to CALVIN and Franka Kitchen benchmarks, there are fewer differences in the performance of the playbook and baselines because there are fewer actions required to complete each single task. In conclusion, we confirm that the playbook can be successfully trained using an offline dataset consisting of demonstrations of single tasks.

