# OpenReview forum: "Playbook: Scalable Discrete Skill Discovery from Unstructured Datasets for Long-Horizon Decision-Making Problems"
_ICLR.cc/2025/Conference — Submitted to ICLR 2025_

### Official Review · Reviewer_4gAe · 2024-10-27

**Soundness:** 3
**Presentation:** 3
**Contribution:** 4
**Rating:** 8
**Confidence:** 2

**Summary:**

This paper proposes a new algorithmic technique for learning a set of reusable robot skills, called "plays", collecting them in an extensible "playbook", and planning using that playbook to complete tasks. The authors extend existing work on learning a set of motion primitives, and add the notion of a play, a mixture/weighting of these primitives. The list of plays and their weights can be learned end-to-end from demonstrations. The set of plays can be extended to learn new plays/primitives from a new dataset. The plays can also be used for beam search and Monte Carlo Tree Search to enable planning in two configurations - single-plan and mixed-plan. The authors show that in simulated robotics benchmarks, their technique can more frequently complete tasks with several subtasks than existing approaches.

**Strengths:**

* The problem being approached is valuable and the playbook concept is a good addition to the literature (significance).

* The technical approach is an impressively-novel combination of many existing techniques, plus the playbook concept, into a full pipeline. (originality)

* The ablation studies performed and progressively-harder experiments do a good job of showing the graceful degradation of the proposed approach and the improvements over prior work. (quality)

* The diagrams are useful (clarity).

**Weaknesses:**

* I found parts of the paper to be unclear to the point of causing me to doubt the technical details of the approach, although I confess that some of this may be due to unfamiliarity with some of the underlying techniques being utilized. See below:

	* I found the parallels to/usage of FOSTER to be confusing. Around line 245, the paper states "compresses the entire model grown to the original model size through the knowledge distillation technique". However, it also says that in FOSTER, "the size of the entire model is increased". Does the model increase in size, or not? A similar confusion happens in the description of the paper's own procesxs, saying around line 256, "the embedding model is reduced to its original network size", but then around line 318, it is stated that all models have to be maintained (i.e. increasing in size). The paper would be improved by clarifying exactly what "model" or list of models is being referred to, and when and if it is actually increasing or decreasing in size and/or expressiveness. This is an important point because the (non)scalability of the approach to (much) larger datasets is an important characteristic of the method, given the problem being solved.

	* the word "model" seems overused - metric model, planning model, etc. Consider replacing the term "metric model" (line 300) and other references to the same concept with a different term.

* As described around line 251, the approach adds the results of the new and existing embedded models. This does not seem like a valid combination, since primitives in MCP are designed to be combined multiplicatively, not additively. The paper would be better if it justified this choice.

**Style Notes**

* The title of section 5.2 would be clearer if the term "mixed play" were hyphenated, right now it is unclear if this is "mixed-play plan" or "mixed play-plan" from the title alone.

**Questions:**

* Regardless of whether or not the model/list of models are increasing in size during compounded training, does the number of primitives/plays also increase monotonically? The paper would seem to suggest this, but I wasn't sure if there was also some distillation/size reduction step being proposed for the plays and primitives.

* Why do the plan weights modify the primitives by exponentiation, and not multiplication? Since MCP is designed for multiplicative combination, it seems that a multiplicative weight might be more numerically stable to train.

* On line 294, it's stated that a Q-network measures the distance between two states, but Q networks require an action as well. Is it a value network, or if it is a Q network, where does the action come from? (in the final state, there is no action, what is done then?)

* Is there a requirement on the length of the demonstrations in the dataset (even if it is not a set length, do all demonstrations have to be the same length)? This seems like a requirement based on equations such as the goal state selection on line 304, but such a requirement doesn't seem to be explicitly stated anywhere. I believe H is the window size and must remain fixed, but this is only inferred and referenced in the appendices, and not in the main paper.

---

### Official Review · Reviewer_hXHG · 2024-11-03

**Soundness:** 2
**Presentation:** 3
**Contribution:** 2
**Rating:** 6
**Confidence:** 3

**Summary:**

This paper proposes a framework to handle skill discovery from offline datasets in a lifelong learning setup. The paper combines multiple existing works: during skill discovery, it combines vector quantization (VQ) and multiplicative compositional policies (MCP), where VQ network selects the skill embedding that applies to MCP primitives. When a new dataset incomes, the paper uses FOSTER to first expand the aforementioned VQ and MCP networks and then distill them back to the original size. Finally, during task learning, the paper uses planning to solve the goal state reaching task.

**Strengths:**

Lifelong skill discovery receive increasing attention these years and thus this paper focuses on an important topic.

Meanwhile, while the paper uses a combination of prior techniques, to the best of my knowledge, the formulation of the skill selection policy (as the weight of MCQ) is novel.

In addtion, despite that there are many component in the proposed lifelong skill discovery framework, the illustrative figures greatly improve clarity and help readers understand the proposed pipeline.

**Weaknesses:**

My main concern is the lack of discussion and comparison with prior works in lifelong imitation learning, such as [1].
I understand that there are many components and design choices in such a complex framework, but at least I would like to see if we replace a component in the proposed work with one from prior work, how it affects the performance. Maybe in this way, we can have a better understanding of what matters in a controlled way. For example,
1. The paper proposes to use MCQ to make action multi-modal, and in the ablative study, shows that using a single primitive decreases the performance. But in practice, many imitation learning work uses gaussian mixture model or diffusion to create multi-modal actions. Which of these two methods perform better?
2. When new datasets income, if there are demonstration that's similar to existing skills, LOTUS will also adjust existing skills with such samples. In contrast, IIUC, the proposed methods don't allow such adaption, and skills that are similar but slightly different to existing ones are likely to be added as new plays. This issue may make the playbook unnecessarily large. One potential support is that, in Fig 6, the use of old plays are quite low in Close Drawer and Move Slider Left. Even though they should share many similar skills with Open Drawer and Move Slider Right (I may be wrong, feel free to let me know if there are any other reasons for these low reusage values).

[1] Wan, Weikang, et al. "Lotus: Continual imitation learning for robot manipulation through unsupervised skill discovery." 2024 IEEE International Conference on Robotics and Automation (ICRA). IEEE, 2024.

**Questions:**

see weakness

---

### Official Review · Reviewer_WDnA · 2024-11-03

**Soundness:** 2
**Presentation:** 3
**Contribution:** 1
**Rating:** 3
**Confidence:** 3

**Summary:**

In this work, the authors present _Playbook_, which is an algorithm for discovering "skills" from offline, unlabeled behavior datasets. The goal of Playbook is to be able to take a large, unlabelled dataset, and learn skills – which can be think of as sub-policies – that can be then chained together to solve longer-horizon decision making. The authors evaluate the method in two simulated setups: the play-kitchen environment according to D4RL setup, and the CALVIN benchmark. They first evaluate the method on a static sized dataset, and then on CALVIN, they evaluate on an extending dataset.

**Strengths:**

The authors introduce a new way of thinking about learning skills from unstructured and unsupervised behavior datasets. In particular, the strengths of the paper are as follows:

1. The authors identify an interesting problem of learning skills from large behavior datasets without having access to labels. For example, a lot of large human behavior datasets could be composed of the demonstrators switching between multiple behavior modes, and being able to identify those modes properly will help learn behaviors more efficiently. The authors target the hierarchical learning problem, which can be relevant with the right algorithm.
2. Similarly, I can see the relevance of incremental learning in such a setup – especially in a life-long manner where relevant data may be coming in sequentially.
3. The algorithmic setup for learning a playbook, at a very high level, makes sense. I appreciate the quantization setup – which makes sense for learning a state-and-goal conditioned policy.
4. Finally, using the planning setup to select a play is also an interesting idea – especially if we have a reasonable model of the world. I think this idea is novel, and I have not seen this used in other works before.

**Weaknesses:**

However, there are many points where the work can improve. Some of the major points being:

1. The primary issue with this work is that it seems very much like a stream-of-consciousness work. The paper starts with a playbook learning method, then it talks about sequential learning of the plays, and finally, talks about the sequental planning with plays. All of these are interesting ideas in itself, and it feels like the paper doesn't give any of the ideas the proper time and explanation.

2. Expanding upon the previous point – just the skill playbook learning without even extensibility – requires 5 different continuous hyperparameters, even without considering the number of plays. Then the authors add the monte carlo tree search planning on top. This whole paper keeps stacking ideas one atop another until there is no scope for understanding what is _the_ contribution for this paper, and how to properly evaluate it.

3. Because the whole paper and the proposed method is so complex, the evaluations seem inadequate – only two sim environments, and on relatively small datasets as well. Evaluation on more environments would be helpful to make a case for this method.

4. The baselines seem inadequate – if the task is goal reaching, why are the evaluations only against offline RL methods, and not any goal conditioned BC methods? Similarly, the authors don't seem to compare against the SOTA methods on CALVIN either.

5. The method doesn't clarify all the assumptions it makes. For example, the planning requires a way to judge the "best final state", or some type of metric on the state space. It's not a trivial assumption, but the paper makes it look like no big deal. Similarly, it's not clear how the authors are using trajectory transformer on the CALVIN environment to learn a trajectory where the observations in CALVIN are RGB images.

6. Finally, the ablation studies need more details – for example, how the number of plays were picked is an important question.

**Questions:**

1. How did the authors pick the number of plays, and also the hyperparameters in eq 5?
2. Are there any issues on training this method stably?

---

### Author Response · Authors · 2024-11-26
**Main Response by Authors**

We sincerely thank all reviewers for dedicating their valuable time to providing us with insightful feedback. It’s clear that each review carefully considered our work, and these reviews were instrumental in helping us make significant revisions to our paper. Below, we provide a summary of the major changes made during the discussion period, along with further explanations of our contributions.

```1. Major Modifications.```

$\textbf{(1)}$ We have revised the overall paper, including texts and figures, to make the explanation clear.

$\textbf{(2)}$ We have added the description and result of the experiment about a playbook with the Gaussian mixture model to Section 6.5 and Table 5.

$\textbf{(3)}$ We have added the description and result of the experiment about a playbook with behavioral cloning to Section 6.5, Table 5, and Appendix I.

$\textbf{(4)}$ We have added the description and result of the experiment for the new benchmark, ManiSkill3 [1], to Appendix I and Table 18.

$\textbf{(5)}$ We have updated the supplementary material (source code) by adding the experiment code for the ManiSkill3 benchmark.

```2. About Contributions.```

As our reviewers mentioned, our paper proposes a discrete skill discovery method by combining previous works into a single pipeline. Importantly, we do not use existing studies as they are; we use previous works by transforming them to form a framework based on various novelties.

$\textbf{(1)}$ We use a discrete skill as a weight for the combination of primitives, which not only expresses a multi-modal action distribution but also makes it scalable.

$\textbf{(2)}$ We propose the information bottleneck objective for effective skill discovery, which helps to project action sequences with similar intentions to the same skill.

$\textbf{(3)}$ We extend the MCP structure to adapt new skills. MCP is not structurally expended in the original paper.

$\textbf{(4)}$ When expanding the skill set, we utilize the class-incremental learning techniques to mitigate the catastrophic forgetting problem, which is possible since we have interpreted the goal-conditioned decision-making problem as a skill classification problem.

$\textbf{(5)}$ We use the playbook to solve the compounded problem, which is a mixture of tasks belonging to different datasets. This is an important problem that skill-learning methods should deal with.

Once again, we deeply appreciate the reviewers who gave us constructive reviews to complement and improve our paper. If you have any questions or comments, please feel free to ask. Thank you.

```Reference```

[1] S. Tao, et al., ManiSkill3: GPU parallelized robotics simulation and rendering for generalizable embodied AI, arXiv, Oct. 2024.

---

### Author Response · Authors · 2024-12-02
**Main Response by Authors #2**

```Dear. Reviewers```

We would like to report further experimental results on the ManiSkill3 benchmark.
We couldn't add these results to the paper because the paper revision deadline has passed, but we would appreciate it if the reviewer could refer to it.
Unlike the previous experiment using state-based observations in ManiSkill3 reported in Appendix K, we trained the playbook and baselines using RGB observations in ManiSkill3.

```1. Demonstration Collection.```
First, we trained RGB observation-based SAC agents to gather demonstrations.
The observation is a 64x64x3 RGB image, and the action represents the position and orientation of the end-effector and gripper status in seven dimensions.
To collect diverse trajectories, we added small-scale uniform random noise to actions.
The target tasks are "PushCube-v1”, "PullCube-v1”, "PokeCube-v1”, and "LiftPegUpright-v1”, which are the same as the previous experiment.
To increase data efficiency and performance, we collected high-quality demonstrations using SAC agents with high task success rates. As a result, we were able to successfully train the playbook with only a small number of demonstrations; 500 successful demonstrations were collected for each task and used for training.
All demonstrations are used as a single offline dataset without any task labels.

```2. Experiment Results.```
We measured the average success rate in 100 episodes with three random seeds for all algorithms.
The results are shown in the table below.

|Task|BC|CQL|IQL|Playbook+BC|
|:-:|-|-|-|-|
|PushCube-v1|0.91$\pm$0.04|0.85$\pm$0.07|0.86$\pm$0.10|0.94$\pm$0.04|
|PullCube-v1|0.86$\pm$0.07|0.76$\pm$0.09|0.87$\pm$0.06|0.95$\pm$0.04|
|PokeCube-v1|0.70$\pm$0.17|0.56$\pm$0.12|0.65$\pm$0.05|0.79$\pm$0.03|
|LiftPegUpright-v1|0.52$\pm$0.03|0.43$\pm$0.14|0.51$\pm$0.11|0.52$\pm$0.03|
|Average|0.75$\pm$0.18|0.65$\pm$0.19|0.72$\pm$0.17|0.80$\pm$0.20|

The playbook+BC achieves the highest average success rate of 0.80 and also the best for each task.
As in the results in Appendix K, the performance gaps between the playbook and baselines in ManiSkill3 are smaller than in CALVIN because the agent solves single tasks rather than multi-tasks in ManiSkill3.
Through this experiment, we show that the playbook can be successfully trained using an offline dataset consisting of RGB single-task demonstrations.
If we have a chance, we will add the results and source code of this experiment to the final version.

We extend our deepest gratitude to the reviewers.
If you have any comments, please feel free to ask. Thank you.

---

### Meta-Review · Area_Chair_PuLG · 2024-12-18

**Metareview:**

The paper presents an approach for learning skills from 'play' datasets of behavior. This is a borderline paper with diverging reviews. Since one of the reviewers hasnt engaged in discussion, I have taken a closer look at the reviews, the paper, and the responses. To summarize, on the positive side, several of the motivating ideas and algorithm is interesting and solidly executed. On the negative side, the approach seems overly complex with many moving parts. There were minor concerns on the limited benchmarks, which I believe are addressed with the additional results on maniskill. Another emerging concern is the lack of real-world experimentation that is present in prior work in the area (e.g. C-BeT from Cui et al. 2023). I believe having such results can significantly strengthen the paper as it can show broader applicability of the ideas in this paper. Overall, given the level of difficulty of experiments present in the paper, it is unclear if the excessive complexity of the algorithm is justified.

**Additional Comments On Reviewer Discussion:**

Some of the concerns raised by the reviewers like the additional benchmarking concerns have been addressed. However, many of the concerns of excessive complexity remain.

---

### Decision · Program_Chairs · 2025-01-22

Reject